# SAGA-Dependent Histone H2Bub1 Deubiquitination Is Essential for Cellular Ubiquitin Balance during Embryonic Development

**DOI:** 10.3390/ijms23137459

**Published:** 2022-07-05

**Authors:** Farrah El-Saafin, Didier Devys, Steven A. Johnsen, Stéphane D. Vincent, László Tora

**Affiliations:** 1Olivia Newton-John Cancer Research Institute, Melbourne 3095, Australia; farrah.el-saafin@onjcri.org.au; 2Institut de Génétique et de Biologie Moléculaire et Cellulaire, 67404 Illkirch, France; devys@igbmc.fr; 3Centre National de la Recherche Scientifique (CNRS), UMR7104, 67404 Illkirch, France; 4Institut National de la Santé et de la Recherche Médicale (INSERM), U1258, 67404 Illkirch, France; 5Université de Strasbourg, 67404 Illkirch, France; 6Robert Bosch Center for Tumor Diseases, 70376 Stuttgart, Germany; steven.johnsen@bosch-health-campus.com

**Keywords:** SAGA (Spt Ada Gcn5 acetyl-transferase), USP22, ATXN7L3, ubiquitin, DUB, histone H2B, monoubiquitylation, *Ubc*, mouse, embryos

## Abstract

Ubiquitin (ub) is a small, highly conserved protein widely expressed in eukaryotic cells. Ubiquitination is a post-translational modification catalyzed by enzymes that activate, conjugate, and ligate ub to proteins. Substrates can be modified either by addition of a single ubiquitin molecule (monoubiquitination), or by conjugation of several ubs (polyubiquitination). Monoubiquitination acts as a signaling mark to control diverse biological processes. The cellular and spatial distribution of ub is determined by the opposing activities of ub ligase enzymes, and deubiquitinases (DUBs), which remove ub from proteins to generate free ub. In mammalian cells, 1–2% of total histone H2B is monoubiquitinated. The SAGA (Spt Ada Gcn5 Acetyl-transferase) is a transcriptional coactivator and its DUB module removes ub from H2Bub1. The mammalian SAGA DUB module has four subunits, ATXN7, ATXN7L3, USP22, and ENY2. *Atxn7l3^−/−^* mouse embryos, lacking DUB activity, have a five-fold increase in H2Bub1 retention, and die at mid-gestation. Interestingly, embryos lacking the ub encoding gene, *Ubc*, have a similar phenotype. Here we provide a current overview of data suggesting that H2Bub1 retention on the chromatin in *Atxn7l3^−/−^* embryos may lead to an imbalance in free ub distribution. Thus, we speculate that ATXN7L3-containing DUBs impact the free cellular ub pool during development.

## 1. Ubiquitin Cellular Dynamics, Abundance and Distribution

Ubiquitin (ub) is a highly conserved protein of 76 amino acids that plays a critical role in cellular homeostasis. Through its interaction with target proteins, either alone (monoubiquitination) or in chains of various structures (polyubiquitination), ub can impact processes including protein turn-over, protein signaling, protein trafficking, translational regulation, DNA damage response, transcriptional elongation and mitosis. Ub, either alone or conjugated to another protein, is abundant, accounting for approximately 0.1–5% of total proteins within cells [1,2,3,4]. In mammalian cells, there are four functional genes encoding ubiquitin: two “stress-inducible” polyubiquitin genes, *Ubb* (4 ub tandem units—3 in primates) and *Ubc* (9 ub tandem units), which are composed of tandem ub coding units; as well as two ub-ribosome hybrid genes, *Uba52* and *Uba80* (also known as *Rps27a*), which encode single ub units fused to a subunit of the ribosome (Figure 1). In mammalian cells, ub has a half-life of approximately 28–31 h [5]. Attachment of ub to target proteins is dictated by E3 ub ligase enzymes, which transfer the ub from the E2 ub conjugating enzyme onto a lysine residue in the target protein, via an isopeptide bond. There are more than 600 E3 ligases encoded in the human genome, resulting in target protein specificity [6]. The isopeptide bond between ub and target proteins can be cleaved via deubiquitinating (DUB) enzymes. There are about 100 DUB enzymes encoded in the human genome, allowing for an additional layer in the complexity of ub signaling [7]. Since DUBs can liberate ub from target proteins, they can impact the abundance of free ub in the cell, which can then be re-used to target other proteins. Therefore, the proportion of conjugated ub, monoubiquitinated or polyubiquitinated onto target proteins, as well as the free ub within the cell, is the result of the opposing activities of the ub ligating and the DUB enzymes. Note however, that the transcription efficiency of ub encoding genes as well as the ratio between *Ubb* and *Ubc* mRNA coding for different numbers of ub units per transcripts, as well as the rate of ub degradation, could also affect the free ub pool and thus, may have effects on the proportion of available ubiquitin. Using quantitative mass spectrometry, the precise abundance of ub in human (HEK293T) and mouse (embryonic fibroblast) cells was measured. In both cell types, ~23% of ub was found to be free, ~65% was found monoubiquitinated to target proteins, and ~11% was found in polyubiquitin chains [8]. Interestingly, human and mouse brain cells had a markedly different distribution of cellular ub compared to human embryonic kidney (HEK293T) cells and mouse embryonic fibroblasts (MEFs), with the majority of ub being unconjugated, and a smaller proportion of ub mono- or poly-ubiquitinated to target proteins [8]. In HEK293T cells and MEFs, almost one quarter of cellular ub was found to be associated with histones [8]. Histone proteins are major targets for ubiquitination, with approximately 5–15% of histone H2A and 1–2% of histone H2B being monoubiquitinated [9,10,11,12]. Histone ubiquitination has important cellular functions that impact chromatin compaction, transcriptional elongation, silencing, and the DNA damage response.

## 2. Roles of H2Bub1 in Chromatin Compaction and Transcription

Monoubiquitination (ub1) of histone H2B at lysine 120 in metazoan cells (lysine 123 in yeast) is an evolutionarily conserved mark [11] that plays an important role in cellular homeostasis. Addition of a bulky ub moiety onto histone H2B has been proposed to create a docking site for specific factors, “readers” that in turn can regulate diverse chromatin related activities [13]. Monoubiquitination of histone H2B was also described to impact the assembly of nucleosomes, and therefore affect chromatin relaxation/compaction [14]. However, the precise mechanisms are unclear, with several studies using different approaches, proposing distinct outcomes for the role of H2Bub1 in chromatin dynamics. On the one side, studies using reconstituted nucleosome arrays have described that monoubiquitination of histone H2B results in disruption of chromatin compaction and promotes an open chromatin state, while other studies using MNase sensitivity assays have found that H2Bub1 stabilizes nucleosomes, thereby promoting chromatin compaction [14,15,16,17]. Nevertheless, the local impact of H2B monoubiquitylation on chromatin compaction/relaxation could be dynamic and context dependent, which is why it is difficult to capture its role in different assays, and/or when analyzing isolated sets of genes in a snapshot of time. Although H2B monoubiquitination has been linked to the regulation of chromatin associated processes, including transcription, transcription elongation, DNA replication, mitosis, and meiosis [18], how this histone modification and the erasure of this mark function is not yet well understood.

In mammalian cells, the E3 ub ligase complex consists of a obligate heterodimeric complex of the ring-finger proteins RNF20 and RNF40, which is required for depositing ub onto histone H2B (Figure 1). Interestingly, while the complex functions as a heterodimer in mammalians (RNF20/40) and in fission yeast (Brl1/2), it appears to be homodimeric in *Drosophila* (Bre1) and budding yeast (Bre1p). The establishment of H2Bub1 by RNF20/40 is directly coupled to active transcription by RNA polymerase II (Pol II) and is promoted by the polymerase associated factor complex 1 (PAF1), and the histone chaperone facilitates chromatin transcription complex (FACT) [19,20,21]. As a result of the RNF20/RNF40 interaction with Pol II during transcription elongation, H2Bub1 is deposited on the gene bodies of actively transcribed genes, and consequently is absent from non-transcribed chromosomal regions [16,22]. This effect is dependent upon the activity of the active component of the positive transcription elongation factor-b (P-TEFb) component CDK9, where CDK9 serves a dual role by increasing the activity of the E2 enzyme UBE2A by phosphorylating Ser120 [23] as well as promoting RNF20/40 recruitment to elongating polymerase through phosphorylation of Ser2 within the C-terminal domain of RNA Pol II [24,25]. Thus, H2Bub1 is a dynamic mark, and repeated cycles of histone H2B ubiquitination and deubiquitylation occur in gene bodies of actively transcribed genes both in yeast and mammalian cells [22,26,27].

The DUB module of the SAGA (Spt-Ada-Gcn5 acetyltransferase) transcriptional coactivator complex is the primary complex responsible for removing ub from histone H2Bub1 in eukaryotes [26,28,29,30]. The metazoan SAGA DUB module is composed of four subunits: ATXN7, which anchors the DUB module to the SAGA core complex, as well as ATXN7L3, ENY2, and the DUB enzyme USP22 [30,31]. In addition, two other USP22-related ubiquitin hydrolases, USP27X or USP51, can also interact with ATXN7L3 and ENY2 to form related DUB modules which do not associate with the SAGA complex, and deubiquitylate H2Bub1 [32]. USP22, USP27X, or USP51 require ATXN7L3 and ENY2 for their full activity and are unable to deubiquitinate H2Bub1 alone. In the absence of ATXN7L3, or ENY2, H2Bub1 accumulates on the genome, indicating that without these proteins, the function of the related USPs (USP22, USP27X and USP51) is compromised [30,32,33,34]. Since H2Bub1 is associated with all transcribed genes, these related DUB enzymes recognize rapidly and specifically H2Bub1, and recycle ub from all active genes, thus creating an equilibrium between Pol II transcription-dependent ub deposition (see below) and the recycling of ub by the three related DUBs.

*ATXN7L3* knock-down in human HeLa cells resulted in the retention of H2Bub1 on the gene bodies of actively transcribed genes, indicating that the ATXN7L3-related DUB activities are directed toward the transcribed region of almost all expressed genes. However, the retention of H2Bub1 on actively transcribed gene bodies in HeLa cells lacking *ATXN7L3* did not impact the transcription of all transcribed genes, as measured by RNA-seq, suggesting that there is no correlation between H2Bub1 removal from gene bodies and transcription [22]. When RNA Pol II transcription activity was blocked in wildtype cells, genome-wide H2Bub1 was lost in a few minutes, indicating that the deposition of ub on histone H2B by RNF20/40 is dependent on Pol II transcription [24,35,36]. In contrast, when Pol II transcription was blocked in both human and mouse cells lacking SAGA DUB activity, H2Bub1 was retained, indicating that the SAGA DUB is responsible for removing ub from H2Bub1 to generate free ub [22]. Our recent results further demonstrate that the impairment of H2Bub1 deubiquitylation does not directly impact transcription initiation and/or elongation, because we observed a massive H2Bub1 retention at almost every expressed gene in both *Atxn7l3^−/−^* mouse embryonic stem cells (mESCs) and MEFs (Figure 2), but in contrast, Pol II and Pol II-Ser2P occupancies were only slightly impacted, and only limited subsets of genes changed expression in both cellular systems [34]. In agreement, in *Saccharomyces cerevisiae H2B-K123R* mutant (which cannot be ubiquitinated), the expression of only a low number of genes was affected [37], further suggesting that ubiquitination and deubiquitination may not have global transcriptional regulatory functions. While genome-wide H2Bub1 deubiquitination does not seem to directly impact global regulation of Pol II transcription, we propose that H2Bub1 deubiquitination plays a vital role in generating free ub, and therefore indirectly impacts many cellular processes that utilize ub (discussed further below).

The deposition of ub on H2B within nucleosomes by RNF20/40 has been implicated in a histone tail crosstalk, as it was shown that monoubiquitination of H2B is a prerequisite for trimethylation of histone 3 lysine 4 (H3K4me3) near the promoter regions and histone 3 lysine 79 (H3K79me3) in the early elongating regions of genes both in yeast and mammalian cells [38,39,40,41,42,43]. More recent studies suggest a specific role for H2Bub1 in promoting spreading of trimethylation (me3) of H3K4 into the gene bodies of cell fate-specific genes [44]. Moreover, the transcription-coupled H2Bub1 deposition has been shown to direct trimethylation of H3K4 and H3K79 through the recruitment of the relevant enzymes, SET1 and DOT1, respectively, establishing a histone cross talk [41,42,43], although some degree of specificity does appear to occur. Interestingly, in *Atxn7l3^−/−^* mESCs or MEFs, H3K4me3 levels did not change, despite spite the fact that in these *Atxn7l3^−/−^* cells, H2Bub1 levels increased dramatically by four to ten-fold, depending on the cell type (Figure 2) [34]. Interestingly, during myogenic cell differentiation an apparent disconnection between the H2Bub1-H3K4me3 crosstalk was described, as differentiated myotubes had undetectable H2Bub1 levels, but H3K4me3 levels did not globally change [45]. These observations are interesting since they suggest that deubiquitination of H2Bub1 is not coupled to the H3K4 trimethylation, or to its demethylation. Therefore, while deposition of ub on histone H2B plays an important role directly in transcriptional elongation, and indirectly through the stimulation of H3K4-methylation, the removal of ub from histone H2Bub1 seems to not directly impact transcription, H3K4-methylation or demethylation (Figure 3). However, the activity of the SAGA-DUB plays an important role in cellular homeostasis, as loss of SAGA-DUB activity has a profound impact on development (see below). We propose that rather than a direct role in transcription, the main impact of impaired H2Bub1 deubiquitination is an imbalance in the free ub pool.

## 3. SAGA/ATXN7L3-Related DUBs as Free Nuclear Ubiquitin “Generators” during Development

The available total cellular ub is dictated by four factors: production of new ub molecules from the ub encoding genes, ub degradation via the proteasome, as well as ub conjugation by ub ligases, and ub liberation via DUB activity (Figure 1). Free ub can circulate between the nucleus and the cytoplasm, and be utilized for several processes. Reduction in the availability of free ub impacts mouse embryonic and postnatal development. Loss of the *Ubc* gene results in a 40% reduction in cellular ubiquitin. *Ubc^−/−^* mutant embryos are viable up to embryonic day E11.5, then they start to die. Half of the *Ubc^−/−^* embryos survive to E13.5, however appear growth delayed with a severe liver hypoplasia [46]. Remarkably, the mid gestation lethality and liver hypoplasia could be rescued by the expression of six exogenous copies of the *Ub-HA* transgene inserted in the *Hprt* locus. In contrast, *Ubb* loss of function does not impact embryonic development, but leads to reduced ub levels in the gonads and in the brain resulting in meiosis impairment in males and females, and neuron degeneration in the hypothalamus and in the retina [47,48,49,50,51,52,53,54]. In both *Ubb^−/−^* and *Ubc^−/−^* mutants, there is no efficient compensation by upregulation of *Ubb* or *Ubc*, *Uba52* and/or *Uba80*. First, UBB and UBC contribute differently to the production of cellular ub pool in different organs due to tissue-specific difference in their expression. While UBB is the main ub contributor in fully grown oocytes, in the testis, and in the brain, UBC (with UBA52) is the main contributor in the liver [46,47]. Second, the minimal up regulation of the expression of the other polyubiquitin-encoding genes, as well as *Uba52* and *Uba80,* is not sufficient to compensate the loss of ub production [46,47]. The very specific phenotypes observed in the liver or in gonads and brain suggest that the compensation is, however, efficient in the other organs and one can predict that the phenotype of double *Ubb^−/−^;Ubc^−/−^* mutants would be much more severe than the single mutants. Altogether, the phenotypes of *Ubb^−/−^* and *Ubc^−/−^* mutants demonstrate that the precise amount of cellular ub is important for embryonic development.

The amount of cellular ubiquitin can also be impacted by DUB activity. Ordinarily, the ATXN7L3-containing DUB modules rapidly remove ub from H2Bub1 [22,32,34], generating free ub. Deletion of *Usp22* coding for the DUB enzyme of SAGA leads to vascularization defects in the placenta and lethality between E11.5 and E14.5, but does not affect the levels of H2Bub1 [34,55]. This strongly suggests that, during embryonic development, the H2Bub1 deubiquitination function of USP22 can be compensated by the activity of other USPs, such as USP27X and USP51 [32]. Deletion of *Atxn7l3* results in retention of ub on histone H2B by around three to five-fold in *Atxn7l3^−/−^* embryos and mESCs, and around eight to ten-fold in *Atxn7l3^−/−^* MEFs [34]. *Atxn7l3^−/−^* embryos are severely growth and developmentally delayed and are dead at E11.5 [34]. Although this phenotype is more severe, it is reminiscent of the *Ubc^−/−^* embryos, which start to die at E11.5 and display strong defects in the proliferation of liver progenitors that cannot be compensated by the other ub encoding genes [46,56]. Considering that H2Bub1 accounts for 1–2% of total H2B in wild type mammalian cells, a three to ten-fold increase in H2Bub1 levels in *Atxn7l3^−/−^* cells could translate to a significant sequestration of cellular ub (i.e., an estimated 10^5^ ub molecules in MEFs), which is likely to impact many cellular processes that normally utilize ub. USP22 was also shown to deubiquitinate H2Aub1 in vitro [28,57]. If the USP22-containing DUB would also deubiqitinate H2Aub1, which represents 5–15% of total ubiquitinated proteins, the described sequestration would probably be underestimated. Further studies will need to be carried out to precisely measure the ub abundance in cells and organisms lacking the ability to recycle ub from histone H2Bub1. Moreover, in addition to their functions as potential epigenetic regulators, USP22-, USP27X-, and USP51-containing DUB modules also have non-histone substrates, including TRF1 [58], cyclin D1 [59], FBP1 [60], SIRT1 [61,62], HES1 [63], SNAIL1 [64], and ZEB1 [65]. The SAGA DUB activity has also been implicated in deubiquitination several other cellular proteins, such as heat HSPA5 (also known as BIP or GRP78), receptor-interacting protein kinase 3 (RIPK3), Mediator complex subunits MED16 and MED24, and the largest subunit of Pol II, RPB1 [66,67,68,69]. All these additional deubiquitination activities may also participate to constitute the free cellular ub pool.

Additional observations support the notion that histone-associated ub is dynamic and helps to regulate the cellular ub balance. When nuclear ubiquitin levels were rapidly depleted by the accumulation of nuclear protein aggregates, associated with a neurodegenerative disease, the ub associated with histones H2A and H2B was massively reduced [70], indicating that the ub is recycled from the chromatin and redirected toward the protein aggregates. In addition, several studies have shown that cells treated with proteasome inhibitors have a rapid depletion of histone-associated ub [71,72] including H2B [73], likely because the accumulation of polyubiquitinated proteins destined for proteasomal degradation sequester ub, resulting in a reduction in free ub. Moreover, most stresses (i.e., oxidative stress, osmotic stress, most chemical stresses) induce a very rapid loss of H2Bub1, and knockdown of ATXN7L3 blocks this loss (S.A.J., unpublished data). Since most of these stresses will likely also induce a temporary block in transcription, the observed effects are probably the result of induced deubiquitination by ATXN7L3-dependent DUBs and decreased new transcription-coupled ubiquitination (Figure 1 and Figure 3). These observations together demonstrate that histone-associated ub is not (only) an epigenetic mark but also a potential important source of free ub during cellular stresses. Therefore, sequestration of ub on histone H2B in *Atxn7l3^−/−^* cells likely create a shortage in the dynamic pool of ub that is normally essential for adaptation to cellular stresses.

In conclusion, we suggest that the SAGA-related DUB modules, in addition to, or in parallel with their roles as epigenetic regulators, may also be involved in constituting the free nuclear ubiquitin pool important during embryonic development and thus play a critical role in cellular homeostasis (Figure 1). This observation is especially important considering that some congenital human developmental diseases are caused by mutations in DUBs [74]. Interestingly, mutations in some DUB enzymes, including *AMSH, USP7, USP9X, OTUD6B, OTUD7A,* and *USP27X* are associated with defects in the developing brain and/or heart [75,76,77,78,79,80,81], and the *Atxn7l3^−/−^* embryos (inactivating USP22, USP51, and USP27X) also have severe heart and brain abnormalities [34]. These results together suggest that these organs may be particularly susceptible to imbalances in ub availability or signaling. Similar cyclic ubiquitination-deubiquitination processes exist in the cytoplasm (Figure 1), which may also contribute to the normal cellular ub homeostasis, and thus establishing and measuring ub balance in the different cellular compartments of the cell should be the next technical challenge to determine how cellular ub balance impacts embryonic development.

Many new therapeutics have been developed that target the ubiquitin pathway, likely impacting the cellular ub balance [82,83]. Further work, using sophisticated technologies capable of measuring ub in its various forms, with tiny amounts of material from embryos, will need to be conducted to determine precisely how the imbalance in ub effects cellular homeostasis during embryonic development, and whether the emerging ub targeting therapeutics can restore the ub balance.

## Figures and Tables

**Figure 1 ijms-23-07459-f001:**
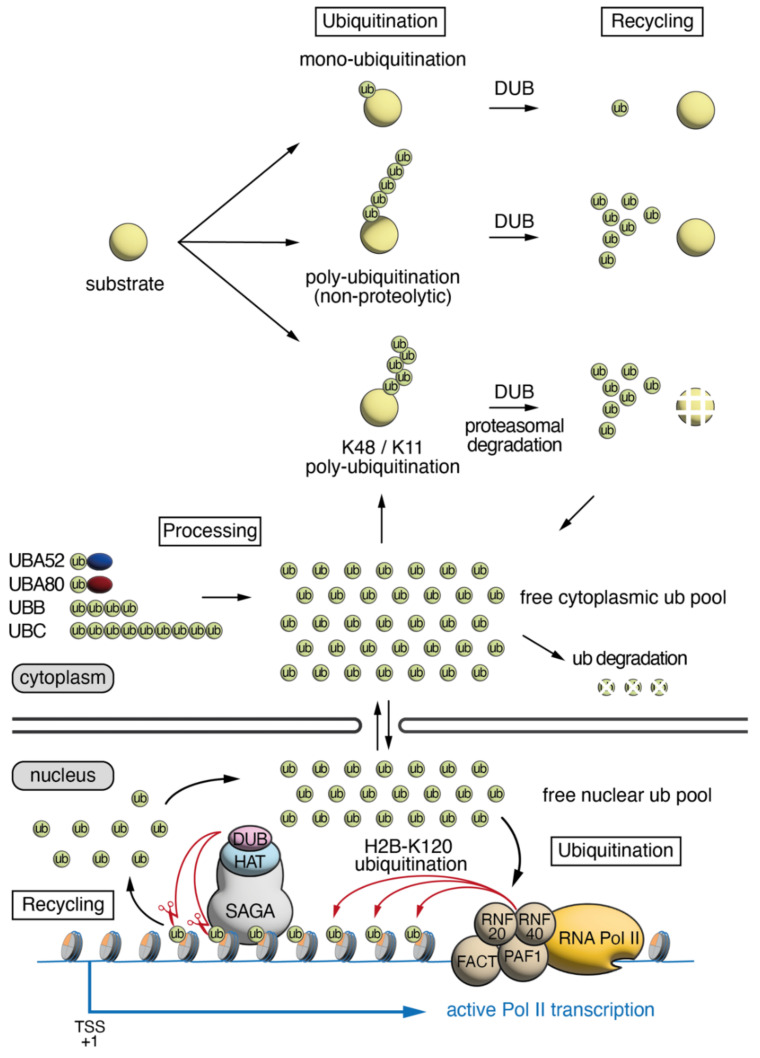
**The cellular dynamics of Ubiquitin.** Four genes encode ub, *Uba52* and *Uba80*, which are ub-ribosome subunit fusion genes (ribosome subunits represented by blue and red ovals), as well as *Ubb* and *Ubc*, which are polyubiquitin genes encoding for tandem ubs. These ub precursor proteins are processed to generate free ub, which can be ligated to substrates either in a mono- or poly-ubiquitin arrangement. Ub can be liberated from its substrate through the activity of DUB enzymes. The resulting free ub can circulate between the nucleus and the cytoplasm, and be utilized for several processes. In the nucleus ub is ligated to histone H2B via the activity of RNF20/40 in a FACT, PAF1, RNA polymerase II (Pol II) transcription-dependent manner. The DUB module of SAGA removes ub from histone H2Bub1 in a transcription-independent manner, contributing to the cellular pool of free ub. TSS: transcription start site.

**Figure 2 ijms-23-07459-f002:**
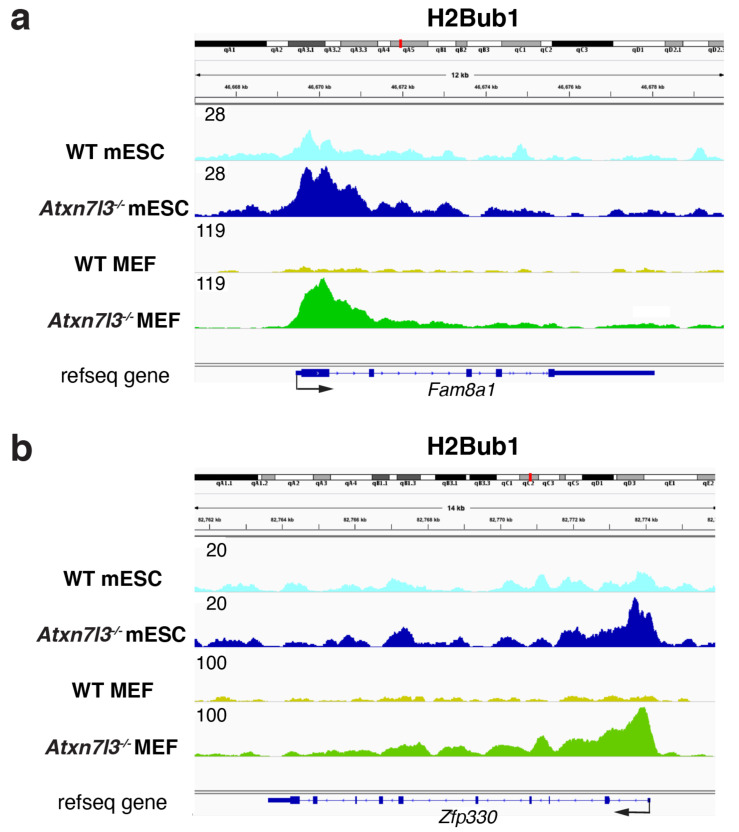
**Histone H2Bub1 levels increase strongly in the gene bodies of both *Atxn7l3^−/−^* mESCs and *Atxn7l3^−/−^* MEFs. **(**a**,**b**) Depletion of the SAGA DUB subunit, ATXN7L3, results in retention of H2Bub1 by around three to five-fold in mESCs, and around eight to ten-fold in MEFs compared to the corresponding wild type (WT) cells. Representative IGV genomic snapshots of H2Bub1-binding profiles are shown at two selected genes (*Fam8a1* and *Zfp330*), re-analyzed from GSE153584 [34]. Direction of the transcription is indicated by arrows. Group scaled tag densities on each gene either in mESCs, or in MEFs, are indicated on the left.

**Figure 3 ijms-23-07459-f003:**
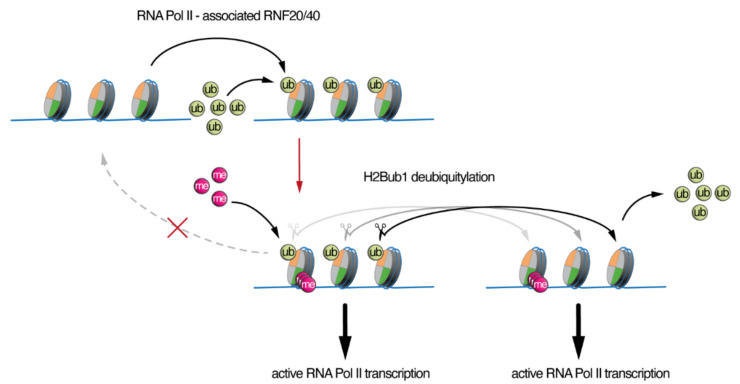
** Asymmetric function of H2Bub1 deposition and deubiquitylation.** H2B ubiquitination is associated with active RNA polymerase II transcription. H2Bub1 facilitates the trimethylation of H3K4 (the histone tail cross-talk is labeled with a red arrow). Removal of monoubiquitin (ub) from histone H2Bub1-containing nucleosomes by the ATXN7L3-dependent SAGA DUB module and the related ATXN7L3-regulated DUBs do not significantly affect the status of H3K4 trimethylation, or RNA polymerase II transcription. However, H2Bub1 deubiquitination does not recreate the original nucleosome state, but instead, could impact the levels of free ub in the cell.

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
