# Peer review of "SAGA-Dependent Histone H2Bub1 Deubiquitination Is Essential for Cellular Ubiquitin Balance during Embryonic Development"

_ijms, 2022, doi:10.3390/ijms23137459_

Round 1

Reviewer 1 Report

The manuscript by Farrah et al. summarizes their research work on ATXN7L3-/- mice and put forward an intriguing hypothesis that the balance of the ubiquitin pool may be the mechanism by which ATXN7L3 knockout affects development. This hypothesis might explain many questions related to USP22 and ATXN7L3 knockout phenotypes. It is a timely review and very well written. I only have a few comments.

 1. In the abstract, the sentence “The cellular distribution of ub is determined by the opposing activities of ub ligase enzymes, and deubiquitinases (DUBs), which remove ub from proteins to generate free ub” is confusing. Cellular distribution means spatial distribution, but I think the authors refer to the distribution of ub pool between conjugate and non-conjugated ub. Please rephrase.

2.  In the abstract and text, “In mammalian cells, 1-2% of total ub is conjugated to histone H2B”. For histone H2B, the ubiquitin modification accounts for 1-2% of total H2B but not total ubiquitin. The same applies to H2A. Please modify the text as needed.

3.  Page 2, “the proportion of conjugated ub, monoubiquitinated or polyubiquitinated onto target proteins, as well as the free ub within the cell, is the result of the opposing activities of the ub ligating and the DUB enzymes.” The transcription and translation efficiency of the ub encoding gene also affects the free ub pool and should have effects on the proportion of ubiquitin pools. Please rephrase.

4. Page 3, paragraph 3, what are the target genes of USP22, USP27X, and USP51? Since H2Bub is associated with all transcribe genes, these deubs should cover all active genes. Please summarize all available ChIP data on these deubs as well as ATXN7L3 to justify the conclusion.

5. Page 4, “Monoubiquitination of histone H2B was also described to impact the assembly of nucleosomes, and therefore affect chromatin relaxation/compaction.” Citation for the sentence. 

6. Page 6, clarify “although some degree of specifies specificity does appear to occur”. 

Author Response

Reviewer 1

Comments and Suggestions for Authors

The manuscript by Farrah et al. summarizes their research work on ATXN7L3-/- mice and put forward an intriguing hypothesis that the balance of the ubiquitin pool may be the mechanism by which ATXN7L3 knockout affects development. This hypothesis might explain many questions related to USP22 and ATXN7L3 knockout phenotypes. It is a timely review and very well written. I only have a few comments.

We were happy to learn that Reviewer 1 liked our review manuscript.

  1. In the abstract, the sentence “The cellular distribution of ub is determined by the opposing activities of ub ligase enzymes, and deubiquitinases (DUBs), which remove ub from proteins to generate free ub” is confusing. Cellular distribution means spatial distribution, but I think the authors refer to the distribution of ub pool between conjugate and non-conjugated ub. Please rephrase.

The point of this sentence is to emphasize the “cellular and spatial” distribution. Thus, we changed the criticized sentence in the Abstract in this direction.

  1. In the abstract and text, “In mammalian cells, 1-2% of total ub is conjugated to histone H2B”. For histone H2B, the ubiquitin modification accounts for 1-2% of total H2B but not total ubiquitin. The same applies to H2A. Please modify the text as needed.

As requested, we have changed these sentences in the Abstract and page 8. In the Abstract now the sentence says: “In mammalian cells, 1-2% of total histone H2B is monoubiquitinated.”

  1. Page 2, “the proportion of conjugated ub, monoubiquitinated or polyubiquitinated onto target proteins, as well as the free ub within the cell, is the result of the opposing activities of the ub ligating and the DUB enzymes.” The transcription and translation efficiency of the ub encoding gene also affects the free ub pool and should have effects on the proportion of ubiquitin pools. Please rephrase.

As requested, this idea has now been incorporated on page 2.

  1. Page 3, paragraph 3, what are the target genes of USP22, USP27X, and USP51? Since H2Bub is associated with all transcribed genes, these deubs should cover all active genes. Please summarize all available ChIP data on these deubs as well as ATXN7L3 to justify the conclusion.

No genome-wide ChIP-seq data is available in mESCs or MEFs concerning USP22, USP27X, and USP51. Nevertheless, as requested, we have added a sentence to better develop our idea and to justify the conclusion. See last sentences on page 4.

  1. Page 4, “Monoubiquitination of histone H2B was also described to impact the assembly of nucleosomes, and therefore affect chromatin relaxation/compaction.” Citation for the sentence. 

The citation has been added.

  1. Page 6, clarify “although some degree of specifies specificity does appear to occur”.

We apologize for the mistake, we corrected it. 

Reviewer 2 Report

Question to authors

Is the nuclear ubiquitin activity associated with ubiquitin activity in cytoplasm?

Are the  similar (as H2Bub1) cyclic processes in cytoplasm? 

This is process coupled with cell division, and ubiquitin i just freelly distributed in the vell?  

Author Response

Reviewer 2

Comments and Suggestions for Authors

Question to authors:

Is the nuclear ubiquitin activity associated with ubiquitin activity in cytoplasm?

We thank the reviewer for this interesting question. To better emphasize this point we amended the legend of Figure 1 and we added a sentence to Part 3 on page 7.

Are the  similar (as H2Bub1) cyclic processes in cytoplasm? 

To speculate about this interesting point, we have added a sentence on page 9 second paragraph of the manuscript.

This is process coupled with cell division, and ubiquitin i just freelly distributed in the vell?  

We apologize, but were not exactly sure what the Reviewer 2 was asking. Thus, respectfully could not answer the question.